# SMAD7 Sustains XIAP Expression and Migration of Colorectal Carcinoma Cells

**DOI:** 10.3390/cancers16132370

**Published:** 2024-06-28

**Authors:** Marco Colella, Andrea Iannucci, Claudia Maresca, Francesco Albano, Carmela Mazzoccoli, Federica Laudisi, Ivan Monteleone, Giovanni Monteleone

**Affiliations:** 1Department of Systems Medicine, University of Rome “Tor Vergata”, 00133 Rome, Italy; marco.colella.bio@gmail.com (M.C.); maresca9595@gmail.com (C.M.); federica.laudisi@uniroma2.it (F.L.); 2Department of Biomedicine and Prevention, University of Rome “Tor Vergata”, 00133 Rome, Italy; andreaiannucci93@gmail.com (A.I.); ivan.monteleone@uniroma2.it (I.M.); 3Department of Biology, Laboratorio di Biologia Delle Cellule Staminali, University of Naples Federico II, 80126 Naples, Italy; francesco.albano@unina.it; 4Laboratory of Preclinical and Translational Research, Centro di Riferimento Oncologico della Basilicata (IRCCS-CROB), 85028 Rionero in Vulture, Italy; carmela.mazzoccoli@crob.it; 5Gastroenterology Unit, Fondazione Policlinico “Tor Vergata”, 00133 Rome, Italy

**Keywords:** SMAD7, STAT3, inflammatory cytokines, F-ACTIN, IAP proteins

## Abstract

**Simple Summary:**

Colorectal cancer (CRC) cells are characterized by high levels of SMAD7, a protein involved in the positive control of growth and survival of cancer cells. This study aims to examine whether SMAD7 is a regulator of CRC cell migration and evaluate the underlying mechanisms. By using specific antisense oligonucleotides, we here show that knockdown of SMAD7 reduces the formation of F-ACTIN filaments and impairs CRC cell migration. Analysis of molecules involved in the control of F-ACTIN formation shows that SMAD7-deficient cells have reduced content of XIAP and this is probably related to the ability of SMAD7 to control the expression of STAT3, a transcription factor that positively regulates XIAP. Finally, we document that, in human CRC samples, there is a positive correlation between SMAD7 expression and XIAP content. Overall, these findings support the role of SMAD7 in the control of F-ACTIN filament formation and migration of CRC cells.

**Abstract:**

The reorganization of the cell cytoskeleton and changes in the content of cell adhesion molecules are crucial during the metastatic spread of tumor cells. Colorectal cancer (CRC) cells express high SMAD7, a protein involved in the control of CRC cell growth. In the present study, we evaluated whether SMAD7 regulates the cytoskeleton reorganization and dynamics in CRC. Knockdown of SMAD7 with a specific antisense oligonucleotide (AS) in HCT116 and DLD1, two human CRC cell lines, reduced the migration rate and the content of F-ACTIN filaments. A gene array, real-time PCR, and Western blotting of SMAD7 AS-treated cells showed a marked down-regulation of the X-linked inhibitor of apoptosis protein (XIAP), a member of the inhibitor of apoptosis family, which has been implicated in cancer cell migration. IL-6 and IL-22, two cytokines that activate STAT3, enhanced XIAP in cancer cells, and such induction was attenuated in SMAD7-deficient cells. Finally, in human CRC, *SMAD7* mRNA correlated with *XIAP* expression. Our data show that *SMAD7* positively regulates *XIAP* expression and migration of CRC cells, and suggest a mechanism by which SMAD7 controls the architecture components of the CRC cell cytoskeleton.

## 1. Introduction

Despite significant advances in the knowledge of colorectal cancer (CRC) pathogenesis and improvements in preventive, diagnostic, and therapeutic approaches, this neoplasia remains one of the major causes of death worldwide [1,2]. This is because CRC diagnosis is formulated when it has already metastasized in more than one-fourth of the patients, and some forms of advanced CRC are resistant to the currently available drugs [3,4]. In this context, it is noteworthy that the vast majority of CRCs have microsatellite stable/proficient mismatch repair (MMR) disease and are poorly responsive to immunotherapy, while CRCs with high-level microsatellite instability (MSI)/deficient (MMR), which account for less than 5% of the stage IV CRCs, is responsive to immunotherapy [5,6]. 

Nearly 5% of CRCs are represented by hereditary syndromes (i.e., adenomatous and hamartomatous polyposis syndrome and Lynch syndrome) and 2% of the cases complicate the natural history of patients with inflammatory bowel diseases, while the vast majority of CRCs occur sporadically [7,8,9,10].

The reorganization of the cell cytoskeleton and changes in the content of cell adhesion molecules are crucial during the metastatic spread of tumor cells [11,12]. The actin microfilaments, microtubules, and intermediate filaments contribute to the metastatic process [12,13]. In eukaryotic cells, actin exists in a globular monomer form, termed G-ACTIN, and a filamentous polymer, termed F-ACTIN [12]. Studies in HCT116 cells, a CRC cell line, showed that ACTIN polymerization and, hence the formation of F-ACTIN, is strictly dependent on the X-linked inhibitor of apoptosis protein (XIAP), a member of the inhibitor of apoptosis (IAP) family [14], which is over-expressed during the cancer progression [15]. The factors/mechanisms regulating XIAP in CRC remain poorly characterized even though indirect evidence suggests that intracellular pathways (e.g., NF-kB and STAT3), which are highly activated in CRC cells and involved in cancer cell metastasis, could up-regulate XIAP expression [16,17,18].

CRC cells are characterized by elevated expression of SMAD7, an intracellular protein, which interacts with the transforming growth factor (TGF)-*β* type I receptor, and suppresses TGF-*β*1-induced phosphorylation of Smad2/Smad3. However, SMAD7 can bind other intracellular proteins and regulate biological functions through a TGF-*β*1-independent manner [19,20]. In CRC, SMAD7 enhances cell growth and survival, and indirect evidence suggests the involvement of SMAD7 in tumor progression and metastatic dissemination [7,21,22,23]. Additionaly, studies in pre-clinical models of CRC showed that systemic administration of a specific SMAD7 antisense oligonucleotide (AS) to mice was sufficient to attenuate tumor growth [19]. Analysis of the mechanisms by which SMAD7 regulates CRC cell behavior revealed that SMAD7 binds the promoter of STAT3, and the AS-induced knockdown of SMAD7 reduces STAT3 mRNA and protein expression in CRC cells, suggesting a role for SMAD7 in the control of STAT3-dependent cancer cell behavior [23]. Based on these findings we hypothesized that SMAD7 might be involved in the invasive potential of CRC cells. Therefore, the present study aimed to evaluate whether SMAD7 regulates CRC cell migration and to explore the underlying mechanisms. 

## 2. Materials and Methods

### 2.1. Cell Culture

All the reagents were from Sigma-Aldrich (Milan, Italy) unless otherwise indicated. The human CRC cell lines HCT116 and DLD1 (American Type Culture Collection, ATCC, Manassas, VA, USA) were cultured in McCoy’s 5A and RPMI 1640 medium, respectively. All media contained 10% fetal bovine serum (FBS) and 1% penicillin/streptomycin (Lonza, Verviers, Belgium). The cells were maintained in a 37 °C, 5% CO_2_, fully humidified incubator and used at a passage number between 10 and 25. In parallel studies, cells were either left untreated or transfected with SMAD7 sense or AS for 24 h and then stimulated with recombinant human IL-6 (30 ng/mL, PeproTech, London, UK) or IL-22 (15 ng/mL, R&D Systems, Minneapolis, MN, USA) for 15–60 min.

### 2.2. Transfection Protocol

HCT116 and DLD1 cells were starved overnight with 0,1% FBS, then washed, and transfected with SMAD7 sense or AS (1.5 µg/mL) for 24 h using Opti-MEM medium and Lipofectamine 3000 reagent (both from Life Technologies, Milan, Italy) according to the manufacturer’s instructions as previously described [19]. Moreover, cells were transfected with commercial STAT3 sense or AS (AZD9150) (both used at 200 µM) (Integrated DNA Technologies, Coralville, IA, USA).

### 2.3. Wound Scratch Assay

The cell migration was evaluated by a wound scratch assay [24]. Briefly, the cells (3 × 10^5^ cells/well) were seeded into a 6-well plate and, after reaching a 100% confluence, were scraped vertically with a sterilized P100 pipette tip (Gilson). Afterward, cells were washed with PBS and transfected with SMAD7 sense or AS. The cells were photographed at 0 and 24 h after transfection using a Zeiss Axiovert 40 CFL inverted microscope (Carl Zeiss, Milan, Italy). Quantitative analysis of the scratch assay was performed by measuring the gap area using the free image-processing software ImageJ, version 1.47 (https://imagej.nih.gov/ij/download.html, 4 May 2024).

### 2.4. Real-Time Migration by xCELLigence System

The 16-well RTCA CIM Plates (Agilent Technologies Inc., Santa Clara, CA, USA) were used to evaluate the cell migration ability in real-time using the xCELLigence System Real-Time Cell Analyzer (ACEA Biosciences, Agilent Technologies Inc., Santa Clara, CA, USA). Experiments were set up according to the manufacturer’s instructions with an uncoated membrane. HCT116 transfected with SMAD7 sense or AS were seeded in serum-free McCoy’s 5A medium in the upper chamber of a CIM-Plate at an optimal number of 3 × 10^3^ cells/well while 10% FBS-supplemented McCoy’s 5A was added in the lower chamber. The cell index (CI) was monitored every 15 min up to 24 h. 

### 2.5. Confocal Microscopy

Briefly, CRC cells were fixed with 4% paraformaldehyde for 15 min and permeabilized with 0.1% TritonX-100 for 5 min at room temperature, blocked for 30 min at room temperature (BSA 1% in 1× PBS), incubated with the TRITC-conjugated phalloidin antibody (1:500 final dilution in methanol (FAK100 Kit) for 1 h, and then stained with DAPI for 1 min. Finally, the slides were washed with PBS and mounted with antifade reagent. The cells were observed under a confocal microscope (Olympus FV1000). Quantitative analysis of the intensity of fluorescence was performed by confocal microscope and normalized on the number of slides (z-stack) and number of cells.

### 2.6. Western Blotting

Cells were lysed on ice in RIPA buffer [10 mM Tris HCl (pH 8.0), 140 mM NaCl, 0.1% SDS, 0.1% Na-deoxycholate, 1% TRITON, 1 mM EDTA, 0.5 mM EGTA, and protease (#04906837001 Roche, Basel, Switzerland) and phosphatase inhibitor] [25]. Lysates were clarified by centrifugation (30 min at 15,000 rpm) and separated on sodium dodecyl sulfate [SDS]–polyacrylamide gel electrophoresis. The membranes were incubated with the following antibodies: SMAD7 [1:1000, #MAB2029 R&D Systems, Minneapolis, MN, USA], XIAP [1:1000, #2042 Cell Signaling Technology, Danvers, MA, USA], and STAT3 [1:1500, #10913 Santa Cruz, Dallas, TX, USA] or β-ACTIN [1:5000 #A544 Sigma] antibody followed by anti-rabbit (1:20000, G21234) or anti-mouse (1:20000, G21D40) secondary antibody conjugated to horseradish peroxidase (Life Technologies, Washoe, NV, USA).

### 2.7. Flow Cytometry

The cells were either left untreated or transfected with SMAD7 sense or AS, or with STAT3 sense or AS (AZD9150), as indicated above. At the end, the cells were collected, washed twice in PBS, stained with FITC-Annexin V (AV, 1:100 final dilution, Immunotools, Friesoythe, Germany) according to the manufacturer’s instructions, and incubated with propidium iodide (PI) (5 mg/mL) for 30 min at 4 °C. The cells were analyzed using flow cytometry Gallios and Kaluza software Version 2.1 (Beckman Coulter Life Sciences, Pasadena, CA, USA).

### 2.8. Real-Time PCR

Total RNA was extracted using the PureLink RNA kit (Thermo Fisher Scientific), and reverse transcription and RT-PCR were performed as previously indicated [23]. cDNA was amplified using the following conditions: denaturation for 1 min at 95 °C; annealing for 30 s at 59 °C for *SMAD7*, *STAT3*, and *B2M*, and 61 °C for *XIAP*; and extension at 72 °C 30 s. To avoid genomic DNA contamination, we designed primer sequences spanning an exon–exon junction. Experiments were performed in triplicates. RNA expression was calculated relative to the *B2M* gene using the ΔΔCt algorithm [26,27]. Primer sequences were as follows: *SMAD7* Fw 5′-GCCCGACTTCTTCATGGTGT-3′, Rev 5′-TGCCGCTCCTTCAGTTTCTT-3′; *STAT3* Fw 5′-GGGAAGAATCACGCCTTCTA-3′, Rev 5′-ATCTGCTGCTTCTCCGTCAC-3′; *XIAP* Fw 5′-CCAAGTGGTAGTCCTGTTTCAG-3′, Rev 5′-GGGATACTTTCCTGTGTCTTCC-3′; *B2M* Fw 5′-GTGGCCTTAGCTGTGCTC-3′, Rev 5′-AGAAAGACCAGTCCTTGCTG-3′.

### 2.9. Transcriptome Analysis

The HCT116 cells were transfected with SMAD7 sense or AS for 24 h. Then, total RNA was extracted using the RNeasy Mini Kit [Qiagen, Hilden, Germany], digested with DNase [Qiagen], and retrotranscribed in complementary DNA. The gene array was performed as previously described [28]. Transcripts were selected based on a fold change value of +1.5 or −1.5, which was generated from the comparison between HCT116 transfected with SMAD7 sense vs HCT116 transfected with SMAD7 AS. Differential Gene Expression (DEG) was analyzed using a bioinformatic free tool, namely “ToppGene Suite” [29].

### 2.10. Gene Expression Profiling Interactive Analysis (GEPIA)

The web portal, GEPIA (http://gepia.cancer-pku.cn/index.html, 4 May 2024), was used to investigate the relationship between *SMAD7* and *XIAP* gene expression in the CRC samples [30,31].

### 2.11. Statistical Analysis

Statistical analyses of the data were performed using GraphPad Prism 6 (GraphPad Software Inc., San Diego, CA, USA). Differences between groups were compared using the Student’s *t*-test and ANOVA (Tukey’s post hoc test). RT-qPCR data were expressed as the fold change of the mean ± standard deviation (SD). *p*-values < 0.05 were considered significant. The correlation between *SMAD7* and *XIAP* mRNA expression in CRC samples as evidenced in the GEPIA database was evaluated using Spearman’s correlation and normalized on *B2M* gene expression.

## 3. Results

### 3.1. SMAD7 Knockdown Reduces CRC Cell Migration

To test the potential role of SMAD7 in tumor invasion, the wound healing assay was performed in HCT116 cells either transfected with SMAD7 sense or AS. The knockdown of SMAD7 (Appendix A) resulted in a great decrease in spontaneous wound healing (Figure 1A). The wounded area in control cells was almost covered by migrating cells after 24 h while there was an evident open area in the SMAD7-deficient cells (Figure 1B). Real-time migration analysis confirmed that SMAD7 knockdown reduced the migration rate of HCT116 cells (Figure 1C). Evaluation of AV/Pi-expressing cells at the same time point revealed that SMAD7 knockdown did not enhance HCT116 cell death (Appendix A). Since F-ACTIN is a powerful driving force for cell motility [32], we evaluated whether SMAD7 knockdown was accompanied by reduced formation of F-ACTIN filaments. Immunofluorescence staining of SMAD7 sense- and AS-transfected HCT116 cells and quantification of the positive cells showed the reduction in F-ACTIN filaments in SMAD7-deficient cells (Figure 1D,E). Similar results were observed in DLD1 cells (Appendix A). 

### 3.2. SMAD7-Deficient CRC Cells Have Reduced Levels of XIAP

To find genes involved in the control of F-ACTIN formation, we conducted a gene array in HCT116 cells transfected with either a SMAD7 sense or AS. Several genes involved in microtubule and cytoskeleton reorganization were differentially modulated in cells transfected with SMAD7 AS (Figure 2A). However, the most down-regulated gene in SMAD7-deficient cells was XIAP (Figure 2A). Time course studies confirmed that down-regulation of *SMAD7* mRNA expression preceded the decline in *XIAP* content (Figure 2B,C). Consistently, the knockdown of SMAD7 was accompanied by a reduced protein expression of XIAP (Figure 2D,E). Similar results were observed in DLD-1 cells (Appendix A–C). Taken together, these findings support the role of SMAD7 in the positive control of XIAP in CRC cells.

### 3.3. XIAP Is Regulated by STAT3 in CRC Cells

CRC cells express high levels of active STAT3 [33,34], and studies in other systems have provided indirect evidence that XIAP could be under the control of STAT3 [35,36]. The JASPAR free web tool (https://jaspar.elixir.no, 4 May 2024) revealed the presence of a potential STAT3 binding site on the XIAP promoter (Appendix A). Consistent with this, the knockdown of STAT3 with AZD9150 in HCT116 cells and DLD1 was followed by a significant down-regulation of XIAP expression at mRNA and protein levels (Figure 3A–C and Appendix A). Evaluation of AV/Pi-expressing cells, at the same time point, revealed the knockdown of STAT3, and AZD9150 in HCT116 cells did not modify the fraction of apoptotic/necrotic cells (Figure 3D).

To further support our hypothesis, HCT116 cells were stimulated with IL-6 or IL-22, two STAT3-activating cytokines that are over-produced in CRC tissue [37,38]. Induction of *XIAP* mRNA was evident as early as 15 min following cytokine stimulation, reached a peak at 30 min, and then declined (Figure 4A,B).

### 3.4. SMAD7-Deficient Cells Fail to Up-Regulate XIAP following IL-6 and IL-22 Stimulation

Since SMAD7 is a direct and positive regulator of STAT3 in CRC cells [23], we next explored the possibility that down-regulation of XIAP in SMAD7-knocked cells was, at least in part, dependent on the SMAD7-mediated control of total STAT3. To explore this issue, HCT116 cells were transfected with SMAD7 sense or AS and then either left unstimulated or stimulated with IL-6 or IL-22. In line with the above data, SMAD7 knockdown was accompanied by a significant down-regulation of XIAP mRNA expression (Figure 4C,D). Moreover, in cells transfected with SMAD7 sense, IL-6 and IL-22 significantly enhanced XIAP mRNA (Figure 4C,D). In contrast, SMAD7-deficient cells failed to up-regulate XIAP following IL-6 stimulation (Figure 4C,D). 

### 3.5. Correlation between SMAD7 and XIAP in Human CRC

By using the GEPIA platform, we showed a positive correlation between *SMAD7* and *XIAP* mRNA expression in human CRC samples (Figure 5).

## 4. Discussion

In recent years, a large body of evidence has been accumulated to support the role of SMAD7 in CRC [39,40]. Specifically, SMAD7 is over-expressed by CRC cells and is supposed to be involved in the positive control of CRC cell growth and survival [19,41]. The present study aimed to further dissect the mechanisms by which SMAD7 regulates CRC cell behavior. Specifically, our goal was to examine whether SMAD7 is a regulator of CRC cell migration and to identify factors/mechanisms involved in such a control. By using a well-standardized model of wound scratch and cell margination, we initially showed that SMAD7-deficient CRC cells have a reduced ability to migrate compared to control cells. Real-time cell assay confirmed the reduced migration rate of CRC cells following SMAD7 knockdown. Notably, analysis of cell death revealed that SMAD7 knockdown did not enhance the fractions of AV/Pi-expressing cells, thus indicating that the reduced cell migration of SMAD7-deficient cells was not secondary to enhanced cell death. Although previous studies have shown that TGF-β1 enhances intestinal barrier integrity and accelerates wound closure in scratch assays [42], it is unlikely the reduced migration of SMAD7-deficient cells is secondary to the suppression of TGF-β1 signaling as our studies were conducted in TGF-β1-unresponsive HCT116 and DLD1 cells. 

During the metastatic process, cancer cells dissociate from the primary tumor, diffuse through blood or lymph vessels, and reach the target organ. These steps require a rearrangement of the cytoskeleton of the cancer cells, which form a range of F-ACTIN-based structures needed for migration and invasion [43,44]. Consistent with this, the diminished migration of SMAD7-deficient cells was associated with a marked reduction in the content of F-ACTIN filaments. A gene array of HCT116 cells transfected with SMAD7 AS aimed at identifying genes involved in the F-ACTIN formation showed a significant down-regulation of XIAP, which was then confirmed by real-time PCR and Western blotting. Moreover, in human CRC samples, there was a positive correlation between *SMAD7* and *XIAP* mRNA expression. 

XIAP is classically known as a regulator of cell death, given its ability to bind and inhibit caspase-9, caspase-3, and caspase-7 [45]. However, XIAP contains a RING domain, which has E3 ligase activity. Therefore, XIAP can degrade, in a caspase-independent manner, several proteins by linking them to ubiquitin molecules [46,47,48]. 

In this context, it has been demonstrated that XIAP-deficient CRC cells exhibit a marked reduction in F-ACTIN polymerization and cytoskeleton formation, and reduced cell migration compared with parental wildtype cells [14]. These data are also consistent with previous studies showing high expression of XIAP in cancer tissues and greater content of XIAP in metastatic specimens than in primary cancers [49,50,51,52]. 

Indirect evidence suggests a potential control of XIAP by STAT3 signaling [53,54]. We have recently shown that SMAD7 binds to the STAT3 promoter and positively regulates STAT3 expression [23]. Therefore, we next explored the possibility that the SMAD7-dependent control of XIAP could be mediated by STAT3. STAT3 knockdown in HCT116 cells was accompanied by a down-regulation of XIAP expression. Moreover, induction of XIAP by IL-6 and IL-22, two STAT3-activating cytokines, occurred in wild-type but not in SMAD7-deficient CRC cells, thus highlighting the involvement of STAT3 in the positive effect of SMAD7 on XIAP expression. However, we cannot exclude the possibility that additional signaling pathways other than STAT3 could contribute to the SMAD7-mediated induction of XIAP. In this context, for example, STAT5 has been associated with the XIAP promoter in vivo, and inhibition of STAT5 in human leukemia virus-transformed T cells by roscovitine, an inhibitor of cyclin-dependent kinases, reduces XIAP expression [55]. Nonetheless, we feel it is unlikely that STAT5 is involved in the modulation of XIAP expression induced by IL-6 and IL-22 in our systems as this transcription factor is mainly activated by IL-2, IL-3, and IL-7 rather than IL-6 and IL-22 [56]. XIAP expression can also be positively regulated by Akt, which phosphorylates XIAP at serine-87, thus protecting it from ubiquitination, degradation, and mitogen-activated protein kinases. However, the involvement of these signaling pathways in the SMAD7-mediated control of XIAP expression remains to be ascertained [57,58,59].

## 5. Conclusions

Our data show that SMAD7 positively regulates XIAP expression and migration of CRC cells. Our data suggest a mechanism by which SMAD7 controls the architecture components of the CRC cell cytoskeleton. Overall, these findings support the role of SMAD7 in colon tumorigenesis and suggest that blockade of this molecule could help manage CRC.

## Figures and Tables

**Figure 1 cancers-16-02370-f001:**
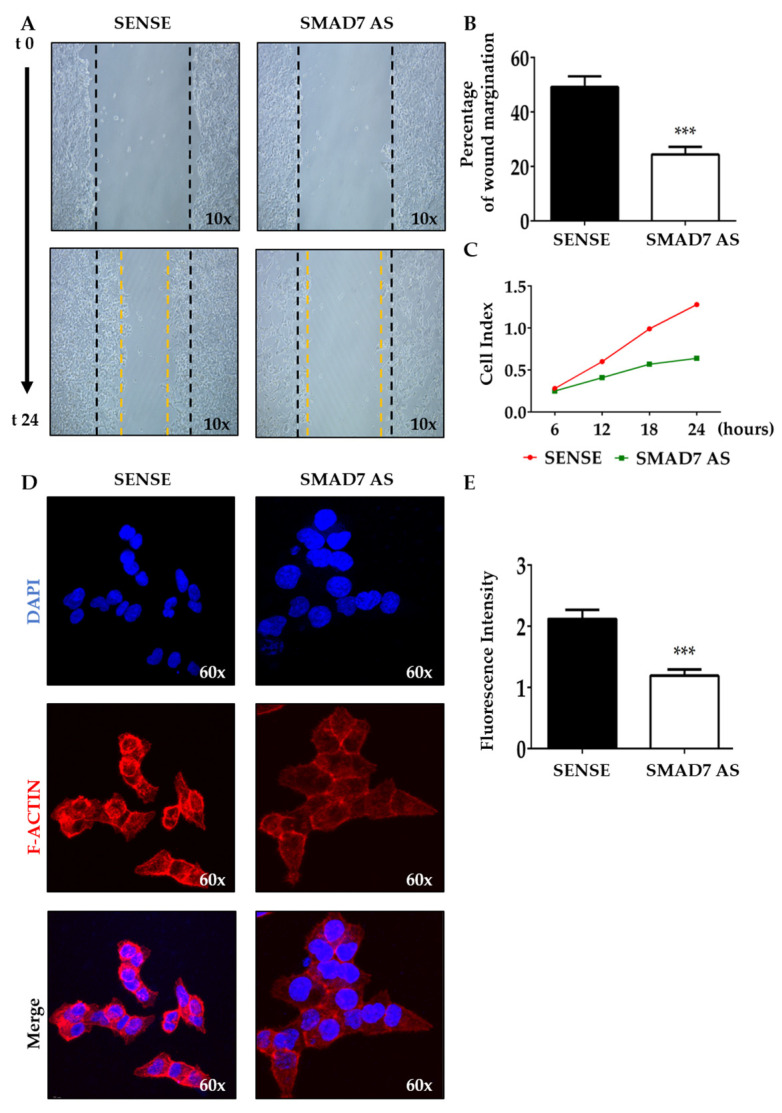
SMAD7 knockdown in HCT116 cells reduces the migration rate and the formation of F-ACTIN filaments. (**A**) The HCT116 cells were transfected with either SMAD7 sense or AS for 24 h. Representative images of cell migration captured at time 0 and 24 h by a phase-contrast microscope (10×). The figure is representative of three separate experiments in which similar results were obtained. (**B**) Quantitative analysis showing the percentage of wound margination at 24 h in comparison to that measured at time 0. The values indicate the mean ± SD; the differences were analyzed using a two-tailed Student’s *t*-test (*** *p* < 0.001). (**C**) The migration rate was monitored in real-time using the xCELLigence system and data indicate the mean values of three independent experiments. (**D**) The HCT116 cells were transfected as above. Representative confocal microscopy images showing F-ACTIN (red) and DAPI (blue) staining (60×). The figure is representative of three separate experiments in which similar results were obtained. (**E**) Quantitative analysis of the fluorescence intensity in cells cultured as above. The values indicate the mean ± SD; the differences were analyzed using a two-tailed Student’s *t*-test (*** *p* < 0.001).

**Figure 2 cancers-16-02370-f002:**
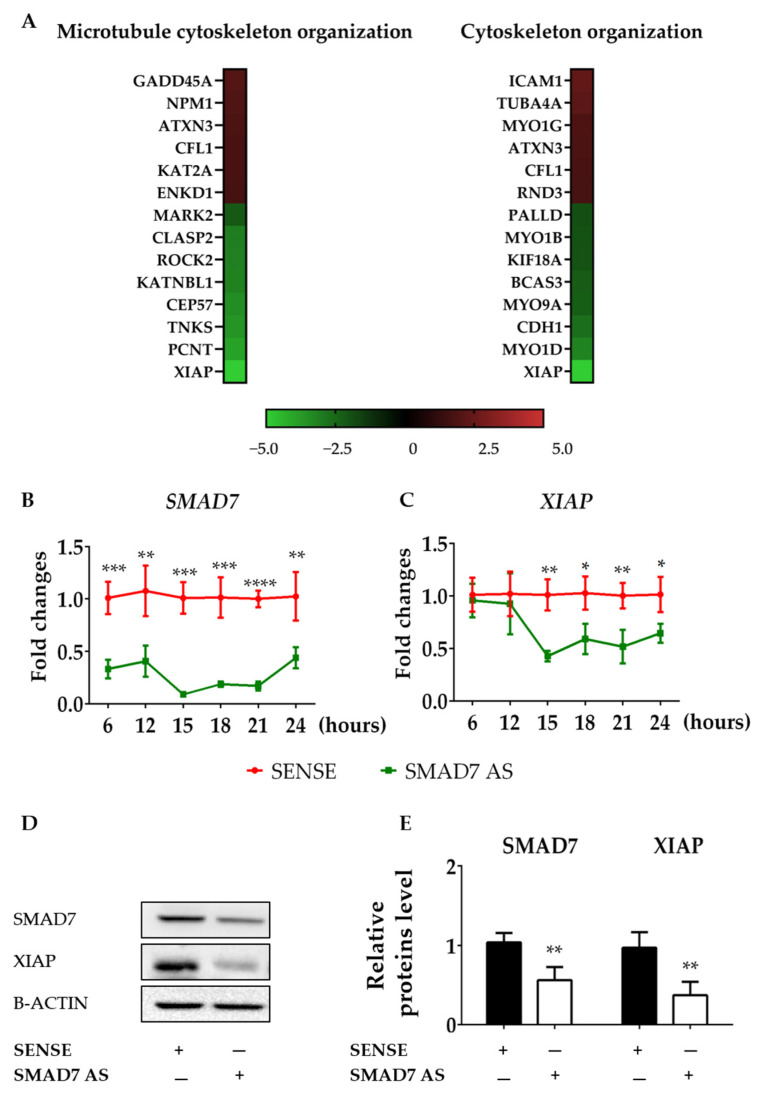
SMAD7 knockdown in HCT116 cells reduces the expression of XIAP. (**A**) Heat map showing microarray-based differential gene expression and log2 [fold change] of genes related to microtubule cytoskeleton organization (GO:0000226) and cytoskeleton organization (GO:0007010) in HCT116 cells transfected with either SMAD7 sense or AS for 24 h. *SMAD7* (**B**) and *XIAP* (**C**) mRNA transcripts were evaluated by real-time polymerase chain reaction. Levels were normalized to B2M. Values show the mean ± SD of three independent experiments. Differences were analyzed using a two-tailed Student’s *t*-test (* *p* < 0.05, ** *p* < 0.01, *** *p* < 0.001, **** *p* < 0.0001). (**D**) Cells were transfected, as indicated in A, and the protein content of SMAD7 and XIAP were analyzed by Western blotting. Panel (**E**) shows the quantitative analysis of SMAD7, XIAP, and B-ACTIN, as evaluated by the densitometry scanning of Western blots. Values indicate the mean ± SD of three independent experiments; differences were analyzed using a two-tailed Student’s *t*-test (** *p* < 0.01). Uncropped Western blot scan be found in Appendix A.

**Figure 3 cancers-16-02370-f003:**
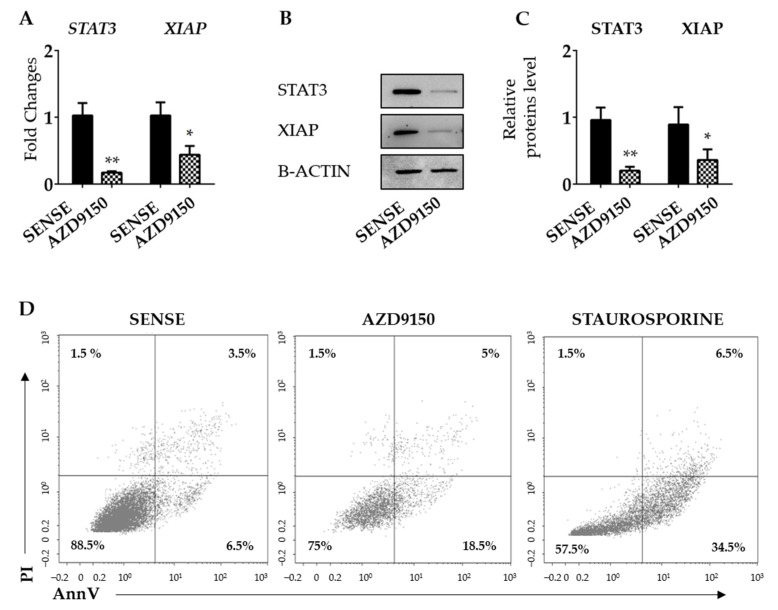
STAT3 knockdown in HCT116 cells reduces the expression of XIAP. (**A**) HCT116 cells were transfected with either STAT3 sense or AS (AZD9150) for 24 h and *STAT3* and *XIAP* mRNA transcripts were evaluated by real-time polymerase chain reaction. Levels were normalized to B2M. Values show the mean ± SD of three independent experiments. Differences were analyzed using a two-tailed Student’s *t*-test (* *p* < 0.05, ** *p* < 0.01). (**B**) Cells were transfected, as indicated in (**A**), and STAT3 and XIAP proteins were analyzed by Western blotting. Panel (**C**) shows the quantitative analysis of STAT3, XIAP, and B-ACTIN, as evaluated by the densitometry scanning of Western blots. Values indicate the mean ± SD of three independent experiments; differences were analyzed using a two-tailed Student’s *t*-test (* *p* < 0.05, ** *p* < 0.01). (**D**) HCT116 cells were transfected as above and the percentages of Annexin V (AV) and/or propidium iodide (Pi)-positive cells were evaluated by flow cytometry. The dot plot is representative of three independent experiments in which similar results were obtained. Values indicate the percentage (%) of positive cells; staurosporine was used as a positive control for 18 h. Uncropped Western blotscan be found in Appendix A.

**Figure 4 cancers-16-02370-f004:**
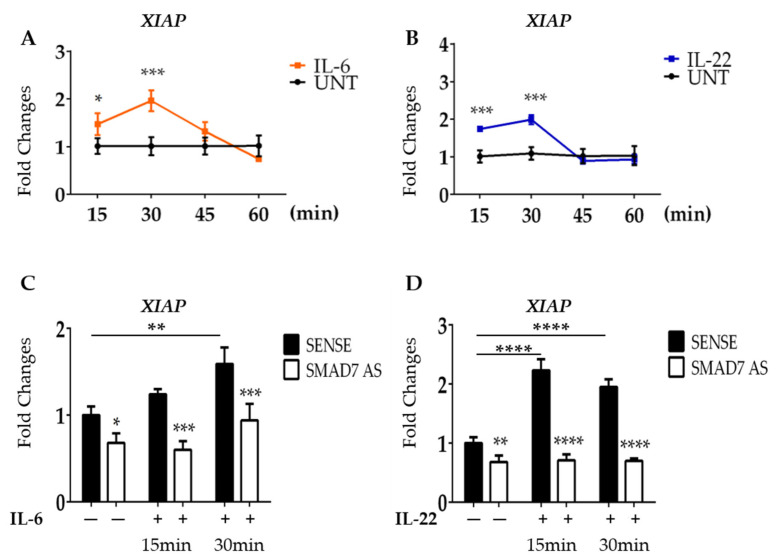
SMAD7 knockdown in HCT116 cells reduces the expression of XIAP following IL-6 and IL-22 stimulation. (**A**,**B**) HCT116 cells were treated with IL-6 (30 ng/mL) or IL-22 (15 ng/mL) at different time points (15–30–45–60 min) and the mRNA of *XIAP* was evaluated by real-time polymerase chain reaction. Levels are normalized to *B2M*. Values show the mean ± SD of three independent experiments. Differences were analyzed using a two-tailed Student’s *t*-test (* *p* < 0.05, *** *p* < 0.001). (**C**,**D**) HCT116 cells were transfected with either SMAD7 sense or AS for 18 h, and then either left untreated or stimulated with IL-6 or Il-22 for 15 or 30 min. *XIAP* mRNA was evaluated by real-time polymerase chain reaction. Levels were normalized to *B2M*. Differences were analyzed using One-way ANOVA (Tukey’s post hoc test) (* *p* < 0.05, ** *p* < 0.01, *** *p* < 0.001, **** *p* < 0.0001).

**Figure 5 cancers-16-02370-f005:**
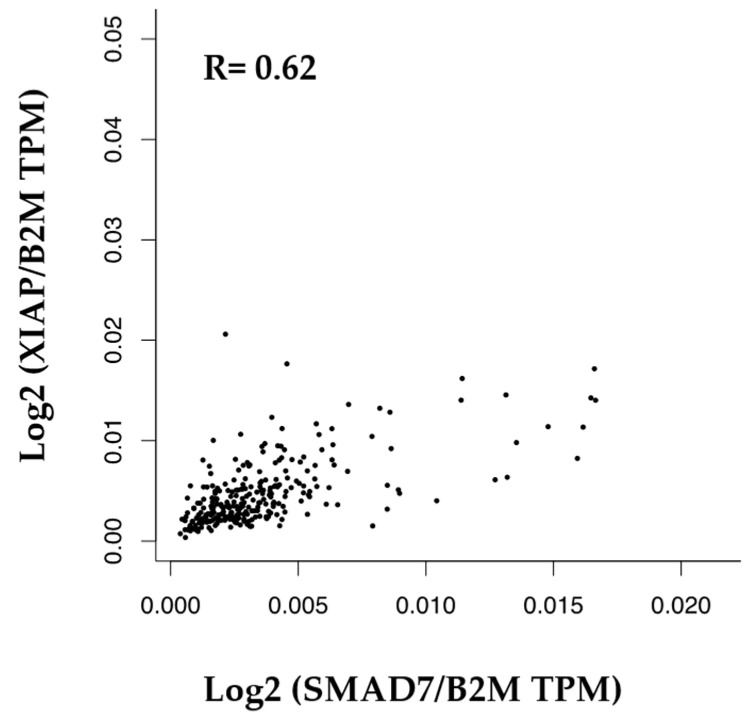
Correlation between the mRNA levels of SMAD7 and XIAP in human CRC.

## Data Availability

The original contributions presented in this study are included in the article; further inquiries can be directed to the corresponding author.

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
