# Peer review of "SMAD7 Sustains XIAP Expression and Migration of Colorectal Carcinoma Cells"

_cancers, 2024, doi:10.3390/cancers16132370_

Round 1
Reviewer 1 Report
Comments and Suggestions for Authors
Dear Editorial Office,
This is an interesting manuscript titled “SMAD7 sustains XIAP expression and migration of colorectal 2 carcinoma cells” and I have some points for improvement:
1. The introduction section is not provided enough especially about the SMAD7 gene. The importance and role of this gene in this study have not been mentioned sufficiently.
2. In the M&M section, cell transfection could be divided or the "cell culture" title should be edited. Because the transfection part is missed when you read the materials and methods.
3. The discussion part is insufficient and it should be extended due to the research's hypothesis and techniques.
Sincerely
Author Response
Reviewer 1:
We would like to thank the reviewer for his/her positive evaluation and helpful comments/suggestions. In response to the minor issues raised by this reviewer:
- The introduction section is not provided enough especially about the SMAD7 gene. The importance and role of this gene in this study have not been mentioned sufficiently
We expanded the introduction section related to SMAD7 gene to better clarify its role in the field.
- In the M&M section, cell transfection could be divided or the "cell culture" title should be edited. Because the transfection part is missed when you read the materials and methods.
We have modified the "cell culture" section and added the information about the “transfection protocol”.
- The discussion part is insufficient and it should be extended due to the research's hypothesis and techniques.
We have modified and expanded the discussion section according to the reveiwer’s suggestion.
Reviewer 2 Report
Comments and Suggestions for Authors
The manuscript entitled “SMAD7 sustains XIAP expression and migration of colorectal 2 carcinoma cells” is a research work focusing on in the role of SMAD7 in colorectal cancer. The authors extend a previous analysis on SMAD7, providing mechanistic insight on how SMAD7 regulates Colorectal cancer migration. They show that SMAD7 regulates XIAP status and migration of colorectal cancer cell lines. Overall, this study is novel with interesting data. However, at present form the manuscript is considered not ready for immediate acceptance as there as some concerns that need to be resolved prior to publication. Below are some issues that could be addressed to strengthen the manuscript:
- Introduction: a brief introduction to colorectal cancer referring to, sporadic, inflammatory-associated and hereditary susceptibility syndromes forms stressing out the distinct molecular alterations and morphological alterations (Refs: PMID: 21530747; PMID: 38279253; PMID: 37001883) is important for non-experts in the field.
- It is well established that TGFβ signaling promotes Epithelial and mesenchymal transition (EMT). As SMAD7 inhibits TGFβ signaling, it is important to address key markers associated with EMT in this setting. Specifically, how SMAD7 knockdown affects E-Cadherin, vimentin (ZEB1 could be added) in HCT116 cells? Along this line how the authors could explain that knockdown of SMAD7 decreases wound migration, while based on the literature it releases TGFβ signaling which in turn promotes EMT?
- As XIAP (PMID: 26232549) modulates caspase-dependent apoptosis, the authors could assess apoptosis upon STAT3 knockdown.
Author Response
Reviewer 2
We would like to thank the reviewer for his/her positive evaluation and helpful comments/suggestions. In response to the minor issues raised by this reviewer:
- Introduction: a brief introduction to colorectal cancer referring to, sporadic, inflammatory-associated and hereditary susceptibility syndromes forms stressing out the distinct molecular alterations and morphological alterations (Refs:PMID: 21530747; PMID: 38279253; PMID: 37001883)is important for non-experts in the field.
We expanded the introduction section to clarify the different aspects related to colorectal cancer.
- It is well established that TGFβsignaling promotes Epithelial and mesenchymal transition (EMT). As SMAD7 inhibits TGFβsignaling, it is important to address key markers associated with EMT in this setting. Specifically, how SMAD7 knockdown affects E-Cadherin, vimentin (ZEB1 could be added) in HCT116 cells? Along this line how the authors could explain that knockdown of SMAD7 decreases wound migration, while based on the literature it releases TGFβ signaling which in turn promotes EMT?
We understand the reviewer’s concern related to TGFβ. However, all the data of the present work were generated using cancer cell lines (i.e. HCT116 and DLD1), which are unresponsive to TGFβ due to TGF-b receptor mutation (see the references below). Indeed, our previous works on SMAD7 in these cell lines showed that SMAD7 controls cancer cell behavior in a TGF-b-independent manner. Therefore, as pointed out -unresponsive CRC cells (1,2).
1) Chiavarina B, et al.; Metastatic colorectal cancer cells maintain the TGFβ program and use TGFBI to fuel angiogenesis. Theranostics. 2021 Jan 1;11(4):1626-1640. doi: 10.7150/thno.51507. PMID: 33408771; PMCID: PMC7778592.
2) Ilyas M, et al.; Transforming growth factor beta stimulation of colorectal cancer cell lines: type II receptor bypass and changes in adhesion molecule expression. Proc Natl Acad Sci U S A. 1999 Mar 16;96(6):3087-91. doi: 10.1073/pnas.96.6.3087. PMID: 10077641; PMCID: PMC15899.
- As XIAP (PMID: 26232549) modulates caspase-dependent apoptosis, the authors could assess apoptosis upon STAT3 knockdown.
We performed the Annexin V/Pi to evaluate apoptosis upon STAT3 knockdown HCT116 cells. The data revealed that knockdown of STAT3 with AZD9150 in HCT116 cells did not modify the fraction of apoptotic/necrotic cells. The results are shown in the figure 3D.
Reviewer 3 Report
Comments and Suggestions for Authors
Dear authors,
Thank you for submitting your manuscript titled "SMAD7 sustains XIAP expression and migration of colorectal carcinoma cells" for consideration. I have carefully reviewed the manuscript and have the following comments and suggestions:
1. Wound healing assay alone may not be sufficient to accurately assess cell migration, as the observed changes could be influenced by cell proliferation. To strengthen the evidence for the role of SMAD7 in regulating CRC cell migration, I would recommend performing additional experiments, such as transwell migration and invasion assays. These assays would provide a more direct evaluation of the cells' migratory and invasive capabilities, independent of proliferative effects.
2. The results figures in the manuscript use uppercase letters (A, B, C, etc.) to label the panels, while the actual figures use lowercase letters (a, b, c, etc.). I suggest ensuring that the labeling in the text matches the figure labeling to maintain consistency and clarity throughout the manuscript.
3. In Figure S1A, the Western blot analysis of SMAD7 and β-actin protein expression appears to have unequal loading between the SENSE and SMAD7 AS groups. To provide a more quantitative assessment of the SMAD7 knockdown, I would recommend performing a densitometric analysis of the Western blot bands and presenting the results in a bar graph format. This would allow for a more accurate and visual representation of the expression differences between the groups.
4. Similarly, for Figure S2A, I suggest presenting the data in a bar graph format to better illustrate the differences in XIAP expression between the SENSE and SMAD7 AS groups. This would improve the clarity and scientific rigor of the presentation.
5. The manuscript would benefit from a more thorough discussion of the potential mechanisms by which SMAD7 regulates XIAP expression and CRC cell migration. While the current discussion touches on the role of STAT3 in this process, a more in-depth analysis of the underlying signaling pathways and their implications would strengthen the overall impact of the study.
6. The language and writing style in the manuscript are generally clear and well-structured. However, I would recommend having the manuscript thoroughly proofread to ensure the consistent use of terminology and the correction of any minor grammatical or typographical errors.
7. The experimental methods appear to be well-designed and appropriately described. However, I would suggest providing more details on the statistical analyses performed, including the specific tests used, the criteria for determining statistical significance, and the handling of any missing data or outliers.
8. The references cited in the manuscript are relevant and up-to-date. However, I would recommend adding a few more recent studies that have investigated the role of SMAD7 in colorectal cancer progression and metastasis to provide a more comprehensive overview of the current literature.
Overall, this is a well-designed study that provides valuable insights into the role of SMAD7 in regulating XIAP expression and CRC cell migration. With the suggested improvements, I believe this manuscript would be a valuable contribution to the field. Please let me know if you have any questions or need further clarification on my comments.
Comments on the Quality of English LanguageMinor editing of English language required,Some statements can be improved
Author Response
Reviewer 3
We would like to thank the reviewer for his/her positive evaluation and helpful comments/suggestions. In response to the issues raised by this reviewer:
- Wound healing assay alone may not be sufficient to accurately assess cell migration, as the observed changes could be influenced by cell proliferation. To strengthen the evidence for the role of SMAD7 in regulating CRC cell migration, I would recommend performing additional experiments, such as transwell migration and invasion assays. These assays would provide a more direct evaluation of the cells' migratory and invasive capabilities, independent of proliferative effects.
We took on board the reviewer’s suggestion and evaluated the migration ability of SMAD7-deficient cells using the Xcelligent assay. These new data confirm that SMAD7 knockdown reduces the HCT116 cell migration. The new results are introduced in the figure 1C in the main text.
- The results figures in the manuscript use uppercase letters (A, B, C, etc.) to label the panels, while the actual figures use lowercase letters (a, b, c, etc.). I suggest ensuring that the labeling in the text matches the figure labeling to maintain consistency and clarity throughout the manuscript.
We corrected all figures accordingly.
3-4. In Figure S1A, the Western blot analysis of SMAD7 and β-actin protein expression appears to have unequal loading between the SENSE and SMAD7 AS groups. To provide a more quantitative assessment of the SMAD7 knockdown, I would recommend performing a densitometric analysis of the Western blot bands and presenting the results in a bar graph format. This would allow for a more accurate and visual representation of the expression differences between the groups.
Similarly, for Figure S2A, I suggest presenting the data in a bar graph format to better illustrate the differences in XIAP expression between the SENSE and SMAD7 AS groups. This would improve the clarity and scientific rigor of the presentation.
We included the densitometric analysis of Western blots.
- The manuscript would benefit from a more thorough discussion of the potential mechanisms by which SMAD7 regulates XIAP expression and CRC cell migration. While the current discussion touches on the role of STAT3 in this process, a more in-depth analysis of the underlying signaling pathways and their implications would strengthen the overall impact of the study.
We focused our discussion on STAT3 just because our data support the role of this transcription factor in the SMAD7-mediated control of cell migration. We had already discussed the possibility that additional signaling pathways other than STAT3 could contribute to the SMAD7-mediated induction of XIAP and acknowledged the role of STAT5 in the control of XIAP promoter. We have now taken on board the reviewer’s comments and discussed the involvement of AKT and MAPK in the positive control of XIAP expression.
- The language and writing style in the manuscript are generally clear and well-structured. However, I would recommend having the manuscript thoroughly proofread to ensure the consistent use of terminology and the correction of any minor grammatical or typographical errors.
We made the requested control of the manuscript.
- The experimental methods appear to be well-designed and appropriately described. However, I would suggest providing more details on the statistical analyses performed, including the specific tests used, the criteria for determining statistical significance, and the handling of any missing data or outliers.
All the statistical analyses are accurately described in the M&M section and figure legends.
- The references cited in the manuscript are relevant and up-to-date. However, I would recommend adding a few more recent studies that have investigated the role of SMAD7 in colorectal cancer progression and metastasis to provide a more comprehensive overview of the current literature.
We included in the main text the reference N°39-40 to clarify the role of SMAD7 in colorectal cancer progression and metastasis.
39)Rosic J, Dragicevic S, Miladinov M, Despotovic J, Bogdanovic A, Krivokapic Z, Nikolic A. SMAD7 and SMAD4 expression in colorectal cancer progression and therapy response. Exp Mol Pathol. 2021 Dec;123:104714. doi: 10.1016/j.yexmp.2021.104714. Epub 2021 Oct 28. PMID: 34717960.
40)Halder, S., Rachakonda, G., Deane, N. et al. Smad7 induces hepatic metastasis in colorectal cancer.Br J Cancer 99, 957–965 (2008). https://doi.org/10.1038/sj.bjc.6604562
Round 2
Reviewer 2 Report
Comments and Suggestions for Authors
The auhtors addressed the comments raised.
Author Response
We would like to thank the reviewer 2 for his/her comments.